# The Impact of Sexual Violence on Quality of Life and Mental Wellbeing in Transgender and Gender-Diverse Adolescents and Young Adults: A Mixed-Methods Approach

**DOI:** 10.3390/healthcare11162281

**Published:** 2023-08-13

**Authors:** Aisa Burgwal, Jara Van Wiele, Joz Motmans

**Affiliations:** 1Transgender Infopunt, Ghent University Hospital, Corneel Heymanslaan 10, 9000 Ghent, Belgium; jaravanwiele@hotmail.com; 2Center for Sexology and Gender, Ghent University Hospital, Corneel Heymanslaan 10, 9000 Ghent, Belgium; joz.motmans@uzgent.be

**Keywords:** transgender, gender-diverse, sexual violence, gender expression, avoidance behavior, quality of life, GHQ-12

## Abstract

Transgender (trans) and gender-diverse (GD) adolescents and young adults have remained largely invisible in health research. Previous research shows worse outcomes in health indicators for trans and GD people, compared to cisgender controls. Research on the impact of sexual violence focuses on mainly cisgender female adult victims. This study assessed the impact of sexual violence on the quality of life (QoL) and mental wellbeing (GHQ-12) among trans and GD adolescents and young adults, while taking into account the possible role of gender nonconformity in sexual violence and mental wellbeing. An online, anonymous survey and interviews/focus groups were conducted between October 2021 and May 2022 in Belgium. Multiple analyses of covariance (ANCOVAs) were used to assess the associations between sexual violence, mental wellbeing, and gender nonconformity, while controlling for different background variables (gender identity, sexual orientation, age, economic vulnerability, etc.). The interviews and focus groups were used to validate associations between variables that were hypothesized as important. The quantitative sample consisted of 110 respondents between 15 and 25 years old, with 30 trans respondents (27.3%) and 80 GD respondents (72.7%). A total of 73.6% reported experiences with sexual violence over the past two years (*n* = 81). The mean QoL score was 5.3/10, and the mean GHQ-12 score was 6.6/12. Sexual violence was not significantly associated with QoL (*p* = 0.157) and only marginally significantly associated with GHQ-12 (*p* = 0.05). Changing one’s physical appearance to conform to gender norms, out of fear of getting attacked, discriminated against, or harassed was significantly associated with QoL (*p* = 0.009) and GHQ-12 (*p* = 0.041). The association between sexual violence and changing one’s physical appearance to conform to gender norms was analyzed, to assess a possible mediation effect of sexual violence on mental wellbeing. No significant association was found (*p* = 0.261). However, the interviews suggest that sexual violence is associated with changing one’s physical appearance, but this association is not limited to only trans and GD victims of sexual violence. Non-victims also adjust their appearance, out of fear of future sexual victimization. Together with the high proportion of sexual violence, as well as the lower average QoL and higher average GHQ-12 scores among trans and GD adolescents and young adults, compared to general population statistics, this highlights the need for policy makers to create more inclusive environments.

## 1. Introduction

Media coverage in recent years shows that incidents of sexual violence against transgender (trans) and gender-diverse (GD) adolescents and young adults remain a reality. At the same time, a large proportion of trans and GD individuals do not officially report sexual violence [1,2]. The visibility of various gender expressions and identities has increased as well. On the one hand, this increased social and political visibility raises the chance of coming into contact with trans and GD individuals, which in turn has a positive effect on attitudes towards this group [3,4,5,6]. On the other hand, there may also be social and personal disadvantages associated with an increased visibility. As trans and GD people become more visible, and feel more comfortable being open about their gender identity, they are also more likely to become victims to negative reactions and even violence [7,8,9,10,11].

Consistent with the Standards of Care 8, we use the term transgender and gender-diverse (trans and GD) to be as broad and inclusive as possible, when describing members of the many diverse communities of people with gender identities or expressions that differ from their sex assigned at birth. The term was chosen with the intention of being as inclusive as possible, and to highlight the many diverse gender identities and expressions among trans and GD people [12].

### 1.1. Mental WellBeing of Trans and Gender-Diverse Individuals

The social impact on a person because of their belonging to a minority group related to sex and/or gender may lead to minority stress [13,14]. The minority stress model suggests that poor physical and/or mental health among sexual minorities can, to a large extent, be explained by stress factors caused by a hostile les-, homo-, and bi- (LGB)phobic culture, often resulting in persistent bullying, discrimination, and victimization [14,15,16]. While the minority stress model was developed with regard to LGB people, research has shown that trans and GD people suffer from gender minority stressors too [17,18,19,20,21,22].

The current study assesses quality of life (QoL) and mental wellbeing (GHQ-12). Various studies show that minority groups, such as trans and GD individuals, score lower on mental wellbeing outcomes than the general population [2,23,24,25,26,27,28]. A few studies have assessed wellbeing within this group, though not solely in a Belgian context, or solely among trans and GD adolescents and young adults. The EU LGBTI II study assessed QoL among European trans and GD adolescents and young adults aged 15–24, where a mean QoL score of 5.2/10 was found, which was significantly lower than that among the QoL of trans and GD respondents aged 25 and older, for whom a mean of 6.1/10 was found (*p* < 0.001) [25]. The score was also lower than the average of 7.6/10 among the general Belgian population aged 18 to 24, measured using the European Quality of Life Survey [26]. General mental wellbeing can be measured using the General Health Questionnaire 12 (GHQ-12, short version). This scale has been validated within the general Dutch-speaking population, where an average Cronbach’s alpha of 0.90 was detected. The scale has also been validated within the Flemish transgender population, where a high Cronbach’s alpha (*α* = 0.91) was found. The Flemish study of Motmans, T’Sjoen and Meier [2] assessed the GHQ-12 score among trans people, regardless of age. The mean GHQ-12 score was 3.9/12, and was significantly higher (*p* < 0.001) than the Belgian mean score of 1.3/12 [27]. Other studies find similar results when the GHQ-12 scores of trans and GD individuals are compared to those of cisgender individuals [23,24,28].

### 1.2. Sexual Violence

The proportional rates of sexual violence among trans and GD adolescents and young adults vary considerably, ranging from 31.7% to 42.0% [2,29]. This variability is largely due to differences in the conceptual and operational definitions of sexual violence, which limit the comparability of existing studies, and the ability to draw conclusions [30,31]. Age also seems to be of importance when assessing sexual violence. Various studies highlight the association between sexual violence and age, with younger people reporting more sexual violence [25,32]. Focusing on Belgium, where the current study was conducted, only one nationally representative study is available that assessed sexual violence among LGBTI+ people [32]. This study found that 68% of the LGBTI+ respondents were confronted with at least one type of sexual victimization in the last year. To allow a comparison with the results of the current study with the study of Keygnaert, De Schrijver, Cismaru Inescu, Schapansky, Nobels, Hahaut, Stappers, De Bauw, Lemonne, Renard, Weewauters, Nisen, Vander Beken and Vandeviver [32], the same definition of sexual violence was used, following the definition provided by the World Health Organization [33] (WHO). The WHO suggests a broad definition of sexual violence, including hands-off and hands-on behaviors, and does not specify the gender of the victim or the perpetrator. Hands-off sexual violence refers to violence without any physical contact between the perpetrator and the victim (e.g., verbal or visual sexual harassment). This type of sexual violence can take place online and offline. Hands-on sexual violence refers to violence where physical contact between the perpetrator and victim is present (e.g., sexual abuse with/without penetration, (attempted) rape).

The current study will focus on sexual violence and its associated factors in the Flemish general trans and GD population aged 15 to 25. By relying on the WHO definition of sexual violence, well-established, existing measures are used to incorporate the proportions of sexual violence, and to avoid a gender bias in the item wording.

### 1.3. Impact of Sexual Violence

The Minority Stress Model by Meyer [13,14] emphasizes that individuals belonging to a minority group face additional stressors that impact their wellbeing. Effective experiences with violence, as well as awareness of existing stigma, cause someone to experience minority stress. The most external and explicit sources of minority stress that trans and GD people can experience are actual experiences with violence (including sexual violence).

The impact of violence on physical and mental health depends on the type of violence, the frequency, the characteristics of the perpetrator, and the characteristics of the victim [8]. Motmans, T’Sjoen and Meier [2] showed that experiencing sexual violence was associated with significantly lower psychological wellbeing (GHQ-12) (*p* = 0.007). The EU Agency for Fundamental Rights [29] also found that trans and GD respondents who indicated that they had experienced sexual or physical violence in the past five years showed a significantly lower satisfaction with life than non-victims (*p* < 0.001). This difference remained significant when only 15–24 year old trans and GD victims of sexual or physical violence were compared with non-victims (*p* < 0.001).

Violence specifically aimed at the minority status of the individual (hate crime) causes an increase in feelings of insecurity and hypervigilance [11,34,35,36,37]. Walters, et al. [38] found that trans and GD people are even more likely to have increased levels of vigilance, vulnerability, and anxiety compared to cisgender LGB people. Moreover, individuals who experience a transphobic incident are more aware of their own stigmatized status than those who have not experienced violence.

### 1.4. Motives for Sexual Violence

There are a number of theories about the motives for committing anti-LGBT+ violence, which also applies to transphobic violence. One of the assumed motives underlying anti-LGBT+ violence is gender nonconformity.

#### Gender Nonconformity

Gender nonconformity is a broad term referring to people who do not behave in a way that conforms to the traditional expectations of their gender, or whose gender expression does not fit neatly into a category [39]. Stigma based on gender identity/expression works in a way wherein only two gender options are considered valid in our society: male and female. All gender options other than male and female are devalued [40,41]. A cisgender identity is as a result ideologically equated with ‘normal’ masculinity and femininity, while a transgender or gender-diverse identity is equated with a transgression of these gender norms [42,43,44,45]. Several studies show that transphobic violence stems from an irrational fear of those who do not conform to cultural gender norms, rather than from being provoked by the minority status of the victim themself [40,41]. For trans and GD individuals, a person may not know whether an individual lives fully in a different gender role, or has undergone gender reassignment treatment. However, trans and GD victims of public violence are often those who are visibly trans or GD people (those individuals who cannot be categorized into a clear male or female categorization, or those who still show clear characteristics of their sex assigned at birth) [2]. Behaving or dressing in a way that, according to social norms, only fits the opposite gender, or does not fit into one of the binary gender roles, increases the chance of violent reactions [7,25,46]. This assumes that those who do not exceed gender norms are less likely to experience violence [7,45]. In addition, a study conducted among people who did not belong to a sexual or gender minority group showed that the perception of a non-heterosexual orientation or a transgender identity also led to violence. This confirms the theory that it is not the trans or GD identity itself that predicts the degree of violence, but rather the degree of gender nonconformity [47]. As a result, violence due to gender nonconformity can lead to trans and GD people adjusting their own behavior in order to avoid being in violent situations again [2,36]. Motmans, T’Sjoen and Meier [2] showed that experiencing a sexual transphobic incident caused 40.4% to avoid certain places/people. The study by the EU Agency for Fundamental Rights [29] also showed that 23.4% of trans and GD respondents often-to-always adjusted their appearance out of fear of getting attacked, threatened, or discriminated against. A significant association was found between sexual/physical violence and physical appearance modification. Trans respondents who reported experiences of sexual or physical violence in the past five years significantly more often changed their appearance, out of fear of becoming a victim (again) (*p* < 0.001).

### 1.5. Research Goals

Various studies have assessed the association between sexual violence and mental wellbeing outcomes [2,25,32,48], and between sexual violence and gender nonconformity [25,42,46,49] among trans and GD people. This article assesses the association between sexual violence and quality of life/mental wellbeing among trans and GD adolescents and young adults (15–25 years of age), and examines the role of gender nonconformity in each of these variables. Based on the literature, it is hypothesized that sexual violence is significantly associated with lower mental health outcomes, even when different socio-demographic background variables are taken into account. Avoidance behavior, especially changing one’s physical appearance out of fear of being attacked, discriminated against, or harassed, will also be taken into account, when assessing wellbeing. This type of behavior is associated with gender nonconformity, as changing one’s appearance often refers to a greater degree of gender conformity. It is hypothesized that changing one’s physical appearance will also be significantly associated with both a lower quality of life, and lower mental wellbeing. There is emerging evidence that, in some situations, tests of mediated effects can be statistically significant when the direct effect of the independent variable (e.g., sexual violence) on the outcome variable (e.g., mental wellbeing) is not statistically significant [50,51,52]. If no direct association between sexual violence and mental wellbeing is found, then an indirect mediation effect through changing one’s physical appearance is expected, suggesting full mediation. In comparison with non-victims, it is hypothesized that sexual violence victims will change their physical appearance significantly more often to be more gender conforming, which leads to a significantly lower mental wellbeing.

## 2. Materials and Methods

This study was conducted through the Ghent University Hospital (UZ Ghent), and was funded by the Flemish Government. In 2020, the Equal Opportunities Service proposed to study the experiences with violence of LGBTI+ people, in order to gather empirical evidence about the experiences of LGBTI+ people across Belgium that could be used to improve policy-making and work with LGBTI+ communities. The current article focuses on trans and GD youth with a range of identities, including trans girls (i.e., individuals assigned male at birth, who identify as female or another feminine identity), trans boys (i.e., individuals assigned female at birth, who identify as male or another masculine identity), and gender-diverse or non-binary youth (i.e., individuals who identify as neither male or female, as both male and female, or as another gender identity that is not congruent with their sex assigned at birth).

### 2.1. Study Design

The results in this article were obtained using a cross-sectional, mixed-method design. Experiences of violence were assessed retrospectively, using an online survey, and in-person interviews/focus groups. The respondents who left their contact details at the end of the survey were followed up for an interview, or for participation in a focus group. The main study focused on the experiences with violence of the entire LGBTI+ group. The current article only focuses on the 15–25 year old trans and GD respondents who participated in the main study.

### 2.2. Data Collection Method

The research has been approved by the Commission for Medical Ethics of the Ghent University Hospital. Participants were recruited through the online survey in the study “Genoeg–Enough–Assez”. The survey was hosted by the online survey platform RedCap, and was accessible between October 2021 and January 2022. The interviews and focus groups were conducted between February 2022 and May 2022. LGBTI+ people aged 15 years or older who had lived in Belgium for the past two years were invited to complete an anonymous survey. A convenience sampling strategy was applied, through which the online survey was promoted via posters, flyers, and online (social media). Participants were recruited at different LGBTI+ and non-LGBTI+ events, such as Belgian prides, parties, in queer cafés, during slam poetry events, etc. The LGBTI+ organizations involved in the study helped to reach out to respondents. After each month of data collection, a preliminary analysis was conducted, to check for representativeness, in comparison to the general Belgian population statistics. Based on the results, extra recruitment occasions were scheduled that were specifically aimed at recruiting older LGBTI+ respondents, respondents with a minority ethnic–cultural background, and intersex respondents. At the end of the online survey, people had the option to leave their contact details if they wished to be invited to an interview/focus group. Anyone identifying with an LGBTI+ label could participate, regardless of whether they had experience with violence.

The current article only focuses on the trans and GD respondents who are living in the Flemish/Brussels Capital Region of Belgium, and who are between 15 and 25 years of age. Participants older than 25 years of age, living in the Walloon Region, or with a cisgender identity, were excluded from the analysis. Participants from the Walloon Region were excluded due to there being too small a sample size (*n* = 5). Those who did not sign the informed consent, or who ended the survey before the start of the questions about experiences with violence, were also excluded from the analysis. Of the 110 respondents that fell within the above-defined group and participated in the survey, 32 respondents left their contact details at the end of the survey. All these respondents were invited to participate in the follow-up qualitative part of the study. An invitation to an interview, or an invitation to participate in a focus group, was predetermined at random. In the end, five respondents agreed to participate, of which four were interviewed, and one participated in a focus group.

As the main study focused on LGBTI+ people in general, the focus group also included four cisgender LGB people. Their data were excluded from the qualitative analyses. For both the survey and the interviews/focus group, informed consent forms were signed. It took on average 30 min to complete the survey, and the interviews took approximately one hour, and were audio-recorded. Participants received a debriefing after the survey and interview/focus group, with the contact information of resources that they could reach out to after participating in the study. Participants did not receive an incentive to participate.

### 2.3. Data Analysis Method

Data analysis of the quantitative survey was performed using SPSS for Windows, v28 [53]. A power analysis was conducted to determine the number of participants in this study [54,55,56,57]. Both quality of life and mental wellbeing were continuous variables, and the errors of both dependent variables did not deviate from normality. A series of univariate analyses (ANCOVAs) were performed, to test the hypothesis of a significant difference in mental wellbeing outcomes between young trans and GD individuals who had experienced SV and those who had not, while controlling for different socio-demographic control variables (sexual orientation, sex assigned at birth, gender identity, region, educational level, work situation, relationship status, ethnic–cultural minority, religious minority, minority due to disability status, age, economic vulnerability, and avoidance behaviors). Avoidance behaviors consist of hiding one’s sexual orientation (for example, not holding hands with a romantic partner in public), and changing one’s appearance, out of fear of being attacked, discriminated against, or harassed. Age, economic vulnerability, and both of the avoidance behaviors were used as continuous control variables during the analyses. A *p* value of <0.05 was considered to be statistically significant. The *p*-value is only reported when the control variable is significantly associated with the outcome variable. To achieve a power of 0.80 and a large effect size (1.2), a sample size of at least 105 is required, to detect a significant model. The transcriptions of the interviews and focus groups were analyzed via NVivo, using a thematic analysis [58]. A semantic approach was used, as opposed to a latent approach, to reduce the subjectivity of the researcher’s judgment, and because the interest of this study is more based on people’s stated experiences, rather than their assumptions. For the current article, qualitative data analysis was mainly used, to support the direction of the hypothesized associations, and to find out why certain hypothesized associations turned out to be insignificant.

### 2.4. Main Outcome Measures

We measured Sexual violence (SV) by asking respondents if they had ever experienced one of the following situations in the past two years (‘No’, ‘Yes, once or twice’, ‘Yes, multiple times’). Those respondents answering ‘Yes’ to one of the 21 items were recoded as having experienced SV over the past two years. The World Health Organization’s (2015) definition of SV was adopted, which includes different forms of sexual harassment without physical contact (hands-off SV), sexual abuse with physical contact but without penetration, and (attempted) rape (hands-on SV). Examples of items are ‘Someone said I was sexually inept, abnormal, unattractive…’, ‘Someone stroked, rubbed, or touched the intimate parts of my body against my will (e.g., breasts, vagina, penis, anus)’, and ‘Someone inserted or tried to insert their penis, finger(s) or object(s) into my vagina or anus against my will’.

Quality of life was measured using Q4 from the Quality of Life Survey [26], which respondents answered, according to a 10-point Likert scale, with how satisfied they were with life right now, ranging from ‘not at all satisfied’ to ‘very satisfied’.

Mental wellbeing was measured using the General Health Questionnaire 12 (GHQ-12, short version), in which 12 questions with four answer options were used to derive a total score on 12, in accordance with the Likert scoring method [59]. An example of a question is ‘Have you lost confidence in yourself?’ (with answer options: ‘Not at all’, ‘Not more than usual’, ‘A little more than usual’, and ‘Much more than usual’). The higher the score, the lower wellbeing, or the higher the severity of psychological problems [60].

Participants were also asked a number of socio-demographic questions. Sexual orientation was measured by asking the respondents how they would describe their sexual orientation, with eight answer options: ‘Heterosexual’, ‘Homosexual’, ‘Lesbian’, ‘Bi+’, ‘Asexual’, ‘Other’, ‘I don’t know’, and ‘I don’t want to say’. Based on the open answer responses to ‘Other’, the open answer ‘Queer’ was frequently mentioned, which led to the creation of an extra category. Respondents answering ‘I don’t want to say’ were recoded as missing for this question. Sex assigned at birth (SAAB) was assessed with one question, asking respondents for their sex assigned at birth: ‘Male’, ‘Female’, ‘Other’, ‘I don’t want to say’. The last two answer options were recoded as missing for this question. Gender identity was measured by asking all respondents how they would describe their gender identity at the current moment. A closed list of self-identification options was presented, from which they were asked to select only one option that fits them best: ‘Man’, ‘Woman’, ‘Gender diverse (genderqueer, non-binary, agender, genderfluid, etc.)’, ‘Other’, and ‘I don’t want to say’. If the SAAB was male and the gender identity woman, or if the SAAB was female and the gender identity was man, the respondent was recoded into the category Transgender. If the gender identity was gender-diverse, the respondent was recoded into Gender diverse (GD). Respondents registered the Region in which they currently lived, with the options ‘Flemish Region’, ‘Brussels Capital Region’, and ‘Walloon Region (including the German-speaking community)’. The last option was recoded to missing, due to there being too small a sample size (*n* = 5). We measured the highest obtained educational level by asking about the highest level of education the participant had completed: ‘Without diploma or primary education’, ‘Lower secondary education, general (3 first years completed)’, ‘Lower secondary education, technical/artistic/vocational (3 first years completed)’, ‘Higher secondary education, general (6 years completed)’, ‘Higher secondary education, technical/artistic (6 years completed)’, ‘Higher secondary education, vocational (6 years completed)’, ‘Higher education: graduate, candidacy, bachelor’, ‘University education: licentiate, postgraduate, master’, ‘Postgraduate’, or ‘Doctorate (PhD)’. The first six options were recoded to ‘basic educational level’, and the last four options to ‘advanced educational level’. Work situation was assessed using a multiple response question about the respondent’s current work situation. Respondents could reply to one or more of the following options: ‘Student/in education’, ‘Unemployed/Looking for work’, ‘Long-term ill/incapacitated for work’, ‘Retired (also early retirement, pre-retirement)’, ‘Responsible for everyday shopping and taking care of the household’, ‘Employed (or temporary leave status)’, ‘I don’t want to say’. This variable was recoded into a binary variable indicating whether or not someone is unemployed or long-term ill and not able to work, or employed/retired/taking care of the household. The last option was recoded to missing for this variable. Current relationship status was measured using one question, for which respondents had to choose one option that fitted them best: ‘Single’, ‘I have a partner/partners but do not live with them’, ‘I am married or living together’, ‘I am divorced and not in a new relationship’, ‘I am a widow/widower and not in a new relationship’, ‘Other’, or ‘I don’t want to say’. This variable was recoded into a binary variable with, on the one hand, all respondents indicating they are single, divorced, or a widower, and, on the other hand, all respondents currently in a relationship. Belonging to a minority group (ethnic–cultural, religious, disability status) was assessed using a question for which respondents had to indicate whether they belonged to the minority group (‘Yes’, ‘No’, ‘I don’t want to say’). For each minority group, respondents were recoded into a binary variable, indicating whether or not they belonged to the specific minority group (‘Yes’ or ‘No). The other respondents were recoded to missing. Age was recoded, after asking the respondent for their birth year. Economic vulnerability was measured using a question about how easily the respondents were able to make ends meet. The answer options ranged on a 6-point Likert scale from ‘Very easy’ to ‘With great difficulty’. Hiding one’s sexual orientation or changing one’s physical appearance out of fear of getting attacked, discriminated against, or harassed, was assessed using a 4-point Likert scale, ranging from ‘never’ to ‘always’.

## 3. Results

### 3.1. Study Sample and Characteristics

A total of 110 trans and GD youth between 15 and 25 years of age were included for data analyses. Four youths did not complete key survey items (e.g., sex assigned at birth or current gender identity), and were therefore not included in the final sample. The respondents’ socio-demographic characteristics are summarized in Table 1. All the variables were dichotomized where possible (except for sexual orientation) and age, economic vulnerability, and both of the avoidance behaviors were used as continuous variables.

Interview data from five trans and GD persons between 15 and 25 years of age were used. Most identified as non-binary (*n* = 4); one as a trans man. Four participants lived in the Flanders Region, and one in the Brussels Capital Region, and all participants were white. The youngest respondent was 19 years of age, and the oldest was 23 years of age.

### 3.2. Proportion of Sexual Violence

In total, 73.6% of the trans and GD adolescents and young adults reported having experienced sexual violence over the past two years (*n* = 81). Of these respondents, all reported hands-off sexual violence, with 60.5% of this group reporting at least one experience with hands-on sexual violence (*n* = 49). No significant difference in experiences with sexual violence was found between trans and GD respondents (*X*^2^(1) = 0.28, *p* = 0.60). Economic vulnerability proved to be significantly associated with sexual violence (*p* = 0.037); trans and GD adolescents and young adults who had more difficulties making ends meet more frequently reported experiences with sexual violence.

### 3.3. Quality of Life (QoL) and Mental Wellbeing (GHQ-12)

The mean QoL for 15–25 year old trans and GD respondents was 5.4/10 (*SD* = 1.93). When comparing the respondents who experienced sexual violence in the past two years with those who did not, no significant difference in quality of life was found between victims and non-victims. However, when analyzing the association with the different socio-demographic background variables, a significant association was found between changing one’s physical appearance out of fear of being attacked, discriminated against, or harassed, and quality of life. Trans and GD adolescents and young adults who indicated that they changed their physical appearance more often had a significant lower quality of life than those who did not. The other background variables were not significantly associated with QoL. See Table 2 for the ANCOVA results.

The Cronbach’s alpha of the GHQ-12 scale within the sample was very good (*α* = 0.90), with no item significantly improving the reliability statistic if deleted. The mean GHQ-12 score for 15–25 year old trans and GD respondents was 6.5/12 (*SD* = 2.47). When comparing the respondents who had experienced sexual violence in the past two years with those who had not, a marginally significant difference in mental wellbeing was found between victims and non-victims. When analyzing the association with the different socio-demographic background variables, a significant association was again found between changing one’s physical appearance out of fear of being attacked, threatened, or harassed, and mental wellbeing. The more trans and GD adolescents and young adults changed their physical appearance out of fear, the higher the score on the GHQ-12, which indicates lower wellbeing or a higher severity of psychological problems. The other background variables were not significantly associated with the GHQ-12 score. See Table 3 for the ANCOVA results.

Due to the significant association between changing one’s physical appearance and both the mental wellbeing outcome variables, the association between sexual violence and changing one’s physical appearance was analyzed, to assess the possibility of an indirect effect of sexual violence on mental wellbeing (mediation). Results showed that trans and GD adolescents and young adults who had reported sexual violence in the past two years changed their physical appearance slightly more often, out of fear of being attacked, discriminated against, or harassed, than trans and GD non-victims. However, this association was not significant (*F*(1,105) = 1.28, *p* = 0.261).

### 3.4. Thematic Analysis of the Interviews and Focus Group

The thematic analysis of the transcripts of the interviews resulted in the identification of seven overarching themes: (1) experiences with sexual violence, (2) being trans or GD is experienced as a risk factor for sexual violence, (3) the negative emotional impact of violence, (4) fear and avoidance, (5) acceptance, (6) ignorance, and (7) mental health struggles not specifically related to violence. Some of these themes suggested a clearer picture of the association between sexual violence, changing one’s physical appearance, and the two outcome variables (quality of life and mental wellbeing).

#### 3.4.1. The Negative Emotional Impact of Violence

The direct impact of (sexual) violence on mental wellbeing is most obvious when taking a look at the emotions that respondents reported immediately after experiencing (sexual) violence. They mentioned feelings of gender dysphoria, loneliness, and inferiority. When asked about how they coped with their experiences, some mentioned that it left a permanent mark.

#### 3.4.2. Fear and Avoidance

The long-term impact mentioned most often was fear of future violence and/or judgement. Many respondents reported trying to avoid violence and/or judgement. There were two main strategies: (1) some changed their expression (e.g., dressing differently to adhere to gender norms), and (2) some changed their behavior. These changes in behavior might have given respondents a sense of security, but they also inhibited them from practicing or receiving (self)care (e.g., stopping treatment for mental health problems, due to verbal violence from staff, or no longer going out for late-night mental-health walks, due to catcalling).

The interviews showed that sexual violence has an impact on how people present themselves to the public. However, this proved to be not only the case with victims of sexual violence. The anticipation and fear of sexual violence also caused trans and GD non-victims to behave and dress in a more gender-conforming way, which validates the non-significant association within the quantitative analyses.

## 4. Discussion

Incidents of sexual violence against trans and GD adolescents and young adults remain a reality, and multiple studies have found high proportions of sexual violence among trans and GD individuals. The proportion of sexual violence within the current study is 73.6%, which is a slightly higher proportion than the results found in the representative study of Keygnaert, De Schrijver, Cismaru Inescu, Schapansky, Nobels, Hahaut, Stappers, De Bauw, Lemonne, Renard, Weewauters, Nisen, Vander Beken and Vandeviver [32] (68%). Various studies highlight the association between sexual violence and age, with younger people reporting more sexual violence [25,32]. This could explain why the proportion is higher among trans and GD adolescents and young adults in the current study. However, The EU Agency for Fundamental Rights [25] showed that trans and GD individuals are at greater risk of experiencing sexual violence than other LGB+ groups. The study of Keygnaert, De Schrijver, Cismaru Inescu, Schapansky, Nobels, Hahaut, Stappers, De Bauw, Lemonne, Renard, Weewauters, Nisen, Vander Beken and Vandeviver [32] addressed sexual violence among LGBTI+ people, and the proportion of trans and GD respondents in the sample was very low (1.7%), which might also explain the lower proportion of sexual violence within this study. Future research assessing experiences with sexual violence in this group, using the same conceptualization of sexual violence, and examining the same age group, will provide more insight into the representativeness of the results.

The mental wellbeing of trans and GD adolescents and young adults is also lower, compared to existing European and Belgian statistics. The mean score for quality of life (QoL) was 5.3/10, which is in line with the results of the EU Agency for Fundamental Rights [29], which showed that trans and GD individuals aged 15–25 had an average QoL of 5.2/10. The results are much lower than the average QoL score in the general Belgian population aged 18–24, which is 7.6/10 [26]. The trans and GD adolescents and young adults within the current study appear to have a much lower mental wellbeing, or a higher severity of psychological problems, in comparison with the general Belgian population. The average GHQ-12 score within the current study was 6.6/12, which is much higher than the average of 1.3/12 in the general Belgian population [27], and even much higher than the average score found by Motmans, T’Sjoen and Meier [2], measured among trans respondents (3.9/12).

When the association between sexual violence and both of the wellbeing scores was assessed, only a marginally significant association between sexual violence and the GHQ-12 score could be found (*p* = 0.50). None of the background variables seemed to be significantly associated with either of the mental wellbeing outcome variables, except for changing one’s appearance out of fear of being attacked, discriminated against, or harassed. Trans and GD adolescents and young adults who indicated that they changed their appearance more often had a significantly lower QoL (*p* = 0.009), and a significantly higher GHQ-12 score (*p* = 0.041). The high proportion of sexual violence, as well as the lower average QoL and higher average GHQ-12 scores, among trans and GD adolescents and young adults, compared to general population statistics, highlights the need for policy makers to create more inclusive environments.

The current study provides some validation of the theory of violence, and gender nonconformity as a motive for violence [2,7,25,40,41,45,46]. In accordance with the Minority Stress Theory, trans and GD individuals develop a fear of experiencing violence, based on awareness of existing stigma and actual experiences with violence [17,18,19,20,21]. In anticipating future violence, trans and GD adolescents and young adults indicated that they would modify their appearance to conform to the prevailing binary gender norms, regardless of whether they had been victims of sexual violence in the past. Based on the interviews, it became clear that trans and GD individuals who adapted their appearance more often did not necessarily experience more violence. They indicated that they adapted their appearance out of fear of future victimization. This also validates the non-significant association between sexual violence and gender nonconformity within the quantitative data analyses (*p* = 0.261).

One of the limitations of the current study is that the sampling strategy may have produced skewed data. A convenience sampling strategy was used to collect data. This means that it cannot be guaranteed that the results obtained are representative of the entire Flemish trans and GD community between the age of 15–25. The survey was mainly distributed using posters, flyers, and social media. The posters and flyers were mainly distributed during LGBTI+ events, and in urban areas. Individuals living in rural areas may not have been reached. Furthermore, the quantitative data were gathered online, so respondents were expected to have digital literacy. Nevertheless, the results provide an indicator of what is going on within the Flemish young trans and GD community.

In the past two decades, a number of studies have assessed experiences with violence among trans and GD people. However, a number of references within this article refer to studies from the 1980s and 1990s. Replication of these studies should be performed, to confirm that the theories behind the motives of violence still apply.

## 5. Conclusions

Sexual violence is a common reality among trans and GD adolescents and young adults. The literature has already indicated that people who challenge gender norms are much more visible, and therefore also much more vulnerable. Observing gender-non-conforming behavior can lead to violent behavior. Trans and gender-diverse people do not always fit the binary male/female framework, meaning that a discrepancy between their appearance and their sex assigned at birth can provoke violence. Therefore, awareness campaigns should pay more attention to themes such as gender identity and gender expression. These themes should be included in various training courses, and broader campaigns on violence.

The present study suggests that it is not sexual violence per se that leads to poorer wellbeing. What appears to be more important is the tendency to adapt one’s appearance to be more gender conforming that leads to a lower QoL and GhQ-12 score. Gender nonconformity appears to be an important factor in mental wellbeing. Trans and GD adolescents and young adults who indicated that they adjusted their appearance more often to conform to the binary gender roles in society, out of fear of getting attacked, discriminated against, or harassed, had significantly lower QoL, and a significantly higher GHQ-21 score. We emphasize that these associations are not limited to sexual violence victims, but appear to be similar for both victims and non-victims. The anticipation of future sexual violence leads people to change their appearance, to be more gender conforming. Future studies should try to disentangle the possible reasons behind the association of gender nonconformity and wellbeing.

## Figures and Tables

**Table 1 healthcare-11-02281-t001:** Socio-demographic characteristics of the trans and gender-diverse sample (*N* = 110).

Variable Name	%	*N*
Gender identity		
Transgender	27.3	30
Gender diverse	72.7	80
Sexual orientation		
Heterosexual	3.6	4
Homosexual	7.3	8
Lesbian	15.5	17
Bi+	51.8	57
Asexual	8.2	9
Queer	11.8	13
I don’t know yet	1.8	2
SAAB		
Female (AFAB)	82.2	88
Male (AMAB)	17.8	19
Region		
Flemish	97.3	107
Brussels Capital	2.7	3
Educational level		
Basic	42.7	47
Advanced	57.3	63
Work situation		
Student	75.5	83
Employed	16.4	18
Taking care of the household	1.8	2
Unemployed	8.2	9
Long-term ill	6.4	7
Relationship status		
Single	63.6	70
In a relationship	36.4	40
Minority status		
Ethnic–cultural	6.5	7
Religious	7.7	8
Disability	25	27
Economic vulnerability		
(Very) easily	40.4	44
Relatively easy/with a little effort	46.8	51
With (a lot of) difficulty	12.8	14
Hiding sexual orientation		
Never	19.8	19
Rarely	43.8	42
Often	34.4	33
Always	2.1	2
Changing physical appearance		
Never	13.1	14
Rarely	50.5	54
Often	31.8	34
Always	4.7	5
Age	M	SD
	20.0	3.0

**Table 2 healthcare-11-02281-t002:** ANCOVA analysis results for quality of life (QoL).

Variable	*B*	*F*	*p*	95% CI
Sexual violence (yes)	−0.58	2.03	0.157	[−1.39; 0.23]
Changing physical appearance	−0.64	7.08	0.009 **	[−1.12; −0.16]

Note: ** *p* < 0.01.

**Table 3 healthcare-11-02281-t003:** ANCOVA analysis results for mental wellbeing (GHQ-12).

Variable	*B*	*F*	*p*	95% CI
Sexual violence (yes)	1.10	3.93	0.05	[−2.2; 0.002]
Changing physical appearance	0.69	4.29	0.041 *	[0.03; 1.34]

Note: * *p* < 0.05.

## Data Availability

The data presented in this study are available on request from the corresponding author. The data are not publicly available due to the fact that the dataset contains personal data.

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
