# Peer review of "The Impact of Sexual Violence on Quality of Life and Mental Wellbeing in Transgender and Gender-Diverse Adolescents and Young Adults: A Mixed-Methods Approach"

_healthcare, 2023, doi:10.3390/healthcare11162281_

Round 1
Reviewer 1 Report (Previous Reviewer 4)
Please indicate further how you selected the 5 interviewees from the 110 survey sample. As per your inclusion criteria for the 110 respondents, you "randomly selected" the 5 interviewees from the 110-participant pool. I recommend the following revisions to be reflected in the revised paper:
Of the 110 respondents that fell within the above-defined group and participated in the survey, four were interviewed, and one participated in a focus group. These 5 participants were randomly selected from the 110-participant survey sample.
Author Response
Response to Reviewer 1 Comments
Point 1: Please indicate further how you selected the 5 interviewees from the 110 survey sample. As per your inclusion criteria for the 110 respondents, you "randomly selected" the 5 interviewees from the 110-participant pool. I recommend the following revisions to be reflected in the revised paper:
“Of the 110 respondents that fell within the above-defined group and participated in the survey, four were interviewed, and one participated in a focus group. These 5 participants were randomly selected from the 110-participant survey sample.”
Response 1: The 5 interviewees were not randomly selected from the 110 respondents to the survey. At the end of the online survey, all participants could leave their contact details for a follow-up interview or participation in a focus group. Before they were contacted, they were randomly assigned to participate in an interview or focus group. In total, 32 trans and gender divers respondents between 15 and 25 years of age left their contact details and were invited to participate in an interview or a focus group, depending on the random assignment. In the end, 5 agreed to participate in the follow-up study (15.6%). In light of this information, we added the following to the manuscript:
“Of the 110 respondents that fell within the above-defined group and participated in the survey, 32 respondents left their contact details at the end of the survey. All these respondents were invited to participate in the follow-up qualitative part of the study. An invitation for an interview or an invitation to participate in a focus group was predetermined at random. In the end, five respondents agreed to participate, of which four were interviewed and one participated in a focus group.”

This manuscript is a resubmission of an earlier submission. The following is a list of the peer review reports and author responses from that submission.
Round 1
Reviewer 1 Report
Dear authors,
It was a pleasure to review this manuscript which seeks to study the impact of sexual violence on the quality of life and mental well-being of transgender and gender diverse adolescents and young adults using a mixed methods approach.
I found the subject very interesting and novel, since unfortunately the problems of minority groups are not sufficiently studied.
With the sole objective of improving the quality of the manuscript, I will allow myself to make a series of small comments:
1. The introduction section seemed correct to me. The concepts that will be covered throughout the investigation and the background are explained, immediately putting the reader in the context of the situation. At the end of this section, the authors clearly state the objective of the research.
2. In the material and methods section, the first thing to explain is the study design. Before the data collection method section.
3. In the material and methods section, it is also necessary to explain in detail how the sample was obtained. It is also necessary to explicitly state what the inclusion and exclusion criteria were.
4. In the main measures section (line 219), the tools used should be explained in more detail. Above all, the tools that are validated speak of their reliability through crombach's alpha, if they are validated for the Belgian population or if some cross-cultural adaptation has had to be made.
5. I think the limitations of the study should be written at the end of the discussion section. I suggest removing them from the conclusion section and putting them in the discussion section.
6. I think it would be necessary to rewrite the conclusions section. The conclusions must support the results and respond to the objectives of the investigation. The conclusions should not include the limitations of the study and do not usually include bibliographic citations. It should be something concise and concrete in response to the objectives. I suggest that the text that was written in the conclusions section be taken to the discussion section and the conclusions be rewritten according to the criteria that I suggest at the beginning of this point.
This was my evaluation.
Thank you
Kind regasrds
Reviewer 2 Report
Dear Authors,
I've found your article strongly significant. Nevertheless, I would suggest just a few adjustments:
-the choice to prefer the term GD instead of TGNC would be anticipated (move up to "Introduction");
- in Paragraph 1.1 Sexual violence (lines 70-73) cut the definition;
-Participants is 2.1 and not 3.1 (Results);
- Finally, "Limitations and future research" as separate paragraph and not in the "Conclusions". Limitations include the often overdated references
I feel confident to suggest editing of the English language, better if done by a mother tongue
Reviewer 3 Report
This paper looks at the experiences of sexual violence among TGD people and their impact on health and well-being. The role of gender non-conformity and the need to change gender expression was considered too. Overall, it was a very interesting paper to read. While this is an important topic to consider, edits need to be considered prior to publication. I wish you all the best with your research and I hope my comments are useful.
Introduction
· In section 1.2. concerned with mental health, there are several statements/claims that are not supported with citations. Please ensure this is the case.
· Page 2: “There are few studies assessing general mental well-being among trans and GD individuals, measured with the General Health Questionnaire 90 12 (GHQ-12, short version).”
Why is this measure seen as superior? Can the authors explain whether mental health has been measured using other tools in this population?
· Structure: The structure of introduction could be edited so it is more logical and concise. Currently the authors start off with a section of sexual violence, then move onto mental health, then back to sexual violence in TGD population and, motivates for these behaviours. I would suggest discussing the community’s vulnerability to mental health in response to transphobia, one aspect of that being sexual violence. This term can then be defined, and the motives of the behaviour considered.
· Gender nonconformity is discussed as a predictor/ mediator. There appears to have been research on this factor and its association with sexual violence. Can the authors explicitly explain how exploring this variable in the current study will add to existing knowledge?
Aim:
· From reading the method, it is evident that qualitative data collection also took place. There is no explicit aim set out for the qualitative data. Please can the authors provide a aim and research questions to it is clear why this data was collected?
Method:
· Design: qual and quant data was used in this study. Please can the authors explain the study design that was adopted?
· Page 4: “four respondents were interviewed and one respondent participated in a focus group”. Please can the authors explain how 1 person took part in a focus group? These typically comprise of 2 or more people.
· Did the research take place online or in person?
· What variables were controlled for in the ANCOVA?
· What method of thematic analysis was used?
Findings:
· Again, I am not clear what variables were controlled for in the ANCOVA. Please make this clear.
· Please provide power analyses.
· Results are presenting text and tables. Both is not needed.
· Further rationale, with citations, needs to be provided to explain why mediation was conducted in light of a non-significant finding
· The qualitative data doesn’t appear to have been analysed appropriately. There are no themes presented and just one quote is used. This data needs to be analysed properly or removed from the MS.
· Conclusions about mediation analysis are made based on the qual data. This isn’t appropriate given the lack of significance. Please remove these statements.
· It would be useful to consider the implications of this research too (policy and practice)
Overall this was good but some editing is required to enhance intended meaning.
Reviewer 4 Report
Introduction: Operationally define/describe or give examples for hands-off and hands-on sexual violence.
Method/Participants: Indicate how the 5 interviewees were chosen from the 110 sample.
Results section: give examples of hands-off and hands-on sexual violence.